# A Comprehensive Survey for Deep-Learning-Based Abnormality Detection in Smart Grids with Multimodal Image Data

**Fangrong Zhou [1], Gang Wen [1], Yi Ma [1], Hao Geng [1], Ran Huang [1], Ling Pei [2], Wenxian Yu [2], Lei Chu [2,\*]** 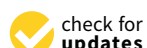 **and Robert Qiu [2]**

1 Electric Power Research Institute, Yunnan Power Grid Co., Ltd., Kunming 650217, China; frzhou.yndw@gmail.com (F.Z.); gwen.yndw@gmail.com (G.W.); daisheng_ds@outlook.com (Y.M.); genghao.yndw@gmail.com (H.G.); rhuang.yndw@gmail.com (R.H.)

2 Shanghai Key Laboratory of Navigation and Location-Based Services, School of Electronic Information and Electrical Engineering, Shanghai Jiao Tong University, Shanghai 200240, China; ling.pei@sjtu.edu.cn (L.P.); wenxianyu@sjtu.edu.cn (W.Y.); rcqiu@sjtu.edu.cn (R.Q.)

\* Correspondence: leochu@sjtu.edu.cn

**Abstract:** In this paper, we provide a comprehensive survey of the recent advances in abnormality detection in smart grids using multimodal image data, which include visible light, infrared, and optical satellite images. The applications in visible light and infrared images, enabling abnormality detection at short range, further include several typical applications in intelligent sensors deployed in smart grids, while optical satellite image data focus on abnormality detection from a large distance. Moreover, the literature in each aspect is organized according to the considered techniques. In addition, several key methodologies and conditions for applying these techniques to abnormality detection are identified to help determine whether to use deep learning and which kind of learning techniques to use. Traditional approaches are also summarized together with their performance comparison with deep-learning-based approaches, based on which the necessity, seen in the surveyed literature, of adopting image-data-based abnormality detection is clarified. Overall, this comprehensive survey categorizes and carefully summarizes insights from representative papers in this field, which will widely benefit practitioners and academic researchers.

**Keywords:** abnormality detection; smart grids; multimodality; image data; intelligent monitoring; statistical inference; deep learning

## 1. Introduction

The reliable management of smart grids plays an important role in every life. Academic researchers and power companies have tried to improve wide-area monitoring, protection, and control (WAMPAC) [1–3] by employing a huge number of PMUs, which enable robust and accurate power state management [4,5]. For example, by the year 2013, more than 2400 PMUs were stationed [1] in major power grids in China. In addition, a vast number of PMUs have been deployed across America. The widespread deployment of PMUs demands precise and reliable monitoring for smart grids [6–8].

An effective way to ensure the stable operation of power grids is to employ the existing measurements from PMUs, which have been extensively studied in the literature. On the other hand, image data sampled from/for power grids can provide a more straightforward means for the management of smart grids, from both short and considerable distances, enabling intelligent monitoring over the air. For example, one can use an infrared/visible image to monitor the status of power lines in an indoor environment, enabling unmanned examination of the status in an all-day standby manner. In addition, one can employ a drone to sample image data for insulator supervision for some outdoor

scenarios, avoiding manual work at heights. Moreover, one can utilize satellite images to perform large-scale abnormality detection. As a result, these multimodal image data play a vital role in abnormality detection in power systems. With the rapid development of advanced data-processing methods, especially deep learning, this role for image data will be further enhanced.

Here, we provide a comprehensive survey of abnormality detection in smart grids using multimodal image data, aimed at providing recent advances and new perspectives to practitioners, academic researchers, and industrial professionals.

## 2. Related Works on Abnormality Detection Based on Multimodal Image Data

Visible Light Images: With the striking advancements in image processing, visible light has been brought to the forefront of research for abnormality detection in smart grids [9–14]. Visible light modality [15–18] means utilizing cameras mounted on various devices to detect transmission line abnormality. Existing methods for collecting visible light images mainly include ground patrol, fixed camera, climbing robot, helicopter, and unmanned aerial vehicle (UAV). Ground patrol is one of the more traditional methods, with humans traveling on foot, which is very slow and labor intensive. In addition, being subject to the influence of extreme weather and harsh terrain, this method is not always possible. Fixed camera is another choice for data collection, but its deployment has a high cost due to the vast number of transmission lines. A climbing robot can achieve high inspection accuracy because of its proximity to the power lines, but its weight may damage the cables. As aerial platforms such as helicopters and UAVs have become available in recent years, their high accuracy and efficiency have made aerial images the central data resource for abnormality detection in smart grids.

**Infrared Images:** Energy utilization [19–22] currently plays a vital role in the development of power systems and smart grids. Therefore, proper monitoring and control of energy consumption is the primary task of the smart grid. For an abnormal power grid, as detected by energy equipment, we summarize three different methods to detect and solve these abnormalities. The equipment in the power grid environment will inevitably have faults and abnormalities in its operation process. For example, pieces of equipment with different functions (such as transformer winding, insulated current transformer, high-voltage disconnector, and watt-hour meter system) have a certain probability of abnormality during operation. Therefore, many detection technologies have been proposed to perform abnormality detection for power grid equipment with different functions. In this paper, we divide the power grid abnormality detection methods into three categories: (1) *Multistage:* A method was introduced by [23] that uses a low-voltage pulse test to check the condition of transformer winding and the infrared control results of electrical equipment; checks the insulated current transformer through thermal and visual control; and measures the acoustic activity of partial discharge in transformer insulation, the gas pressure in the shell, and the data of electrical tests to comprehensively analyze whether there are abnormalities [24,25] in electrical equipment. (2) *Multi-sensor:* A method was introduced by [26] that uses an autonomous, wireless multi-variable intelligent sensor [27–31], which can collect and detect the position of abnormal remote-operated high-voltage disconnectors and use the integrated infrared equipment to detect the temperature data of abnormal switches. (3) *Multi-output:* The methods in [32–34] use an intelligent watt-hour meter based on the Internet of things (IoT) [35–38], which is proposed to properly monitor and control the energy consumption of the watt-hour meter system. The system can accurately control and calculate the energy consumption and upload it to the cloud. It can also accurately detect an abnormal watt-hour meter system, which is helpful in detecting power theft.

Visible light/infrared images enable a solution at a distance of several meters, while optical satellite images can be obtained at a distance of thousands of miles.

Optical Satellite Images: Optical satellite images are better understood by humans because they are in the visible light band. Therefore, by understanding the content of the image, we can deal with various abnormal detection problems in the power grid

environment. For example, Moeller et al. [9] detect towers and conductors by using very high-resolution satellite images. Furthermore, Bernstein et al. [39] proposed an object recognition system that could be used for fast electric power line damage detection after a storm. Due to the limited resolution of satellite optical images, they are not suitable for some more detailed abnormality detection tasks. In recent research, optical satellite imagery has often been used to monitor vegetation cover and the hidden dangers of tree barriers in power grid abnormality detection. The transmission line tree barrier means that the distance between the tree and the wire is constantly being reduced during the growth of the tree, which causes the electric field on the wire surface to be distorted. The strength of the electric field formed between the wire and the tree increases, and this eventually causes the line to flash over, triggering a power outage accident.

This survey paper will be restricted to image-data-based abnormality detection, which is a core technique in WAMPAC for smart grids. For the abnormality detection problem, with the introduction given in [5], the related methods can be broadly categorized into three classes: model-driven [4,5], data-driven [6,7,40–44], and machine-learning-based [45–47]. The model-driven solutions provide a rigorous analysis that is based on solving power flows equations [48]. In contrast, the data-driven ones provide sound solutions based on the statistics extracted from a large amount of data [49–51]. Consider the wide placement of PMUs, this work will focus on the data-driven abnormal detection methods. Interested readers are referred to the works of [1,52–55] for comprehensive reviews and comparisons.

It is worth pointing out that excellent survey works exist in the literature. For example, ref. [56] comprehensively introduced abnormality detection using data-mining techniques. In addition, abnormality detection based on time-series data has been presented by [57]. Moreover, deep-learning-based abnormality detection was described in [58,59]. However, none of them present a comprehensive analysis of deep-learning-based abnormality detection using multimodal image data.

The main contributions that distinguish our survey from those already in the literature [56–60] are the following:

(i) To the best of our knowledge, this is the first survey paper on abnormality detection in smart grids using multimodal image data, which can provide solution examples both in the air and over the air.

(ii) This survey paper is formatted in a comprehensive manner so that interested readers can obtain a wide range of knowledge, which may help them in developing their own ideas in the research area of abnormality detection in smart grids.

(iii) This survey compares both the methods and application scenarios so that interested readers will understand the cons and pros of the employed/compared methods or data quickly, further serving as a road map for academic researchers to start their own works and avoid duplication.

The remainder of this paper is organized as follows. First, Section 2 and Section 3 will introduce image-data-based abnormality detection over short distances. We then look at abnormality detection over large distances in Section 4. Furthermore, we provide the related methodologies in Section 5. Lastly, Section 6 concludes this work.

## 3. Abnormal Detection with Images over Short Distances

### 3.1. Visible Light

Common power line abnormality detection tasks for which visible light can be employed include self-blast glass insulator location, icing detection and measurement, and vegetation encroachment monitoring. We summarize the advantages and disadvantages of these methods in Table 1. Typically, existing approaches for abnormality detection can be broadly classified into two groups: classical methods based on machine learning and conventional image processing, and deep learning methods based on convolutional neural networks (CNN). In this section, we focus on reviewing the above-mentioned methods in smart grids under the visible light modality.

**Table 1.** Visible-light-based abnormality detection.

| Scenario | Objective | Technique Taxonomy | Technique | Literature |
|---|---|---|---|---|
| Self-blast glass insulator location | Locate and identify self-blast glass insulator | Classical methods | MSMF descriptor to extract features, k-means algorithm to generate visual vocabulary of insulator based on local features and spatial orders, coarse-to-fine matching strategy for identification | [61] |
| | | | Texture segmentation algorithm based on PCA and global minimization active contour model (GMAC) | [62] |
| | Deal with the defect detection problems of twin insulator strings | | Segmented from complex backgrounds based on color features, self-shattering based on spatial features of connected regions | [63] |
| | | | Maximum interclass variance method (OTSU) for segmentation, local binary pattern histograms for self-blast detection | [64] |
| | Locate and identify self-blast glass insulator | | SVM and discrete orthogonal S transform (DOST) | [65] |
| | | | Wavelets analysis and SVM | [66] |
| | | | Wavelets analysis combined with hidden Markov model (HMM) | [67] |
| | | Deep learning methods | Orientation angle detection and binary shape prior knowledge (OAD-BSPK), AlexNet, and SVM | [46] |
| | | | Faster R-CNN and U-net | [68] |
| | | | Mask R-CNN | [69] |
| | | | Four-operation data augmentation method, cascaded CNN architecture | [70] |
| Icing detection and measurement | Analyzed three kinds of edge detection methods | Classical methods | Canny operator, Sobel operator, and adaptive weighted Sobel operator | [71] |
| | Automatically detect icing and estimate ice thickness of the transmission lines | | A scheme to improve edge detection accuracy | [72] |
| | Measured the distance and level angle of the ice | | Photogrammetry method based on a laser range finder and inertial measurement unit (IMU) | [73] |
| Vegetation encroachment monitoring | Locating the height of trees, distance between trees and poles, and distance between dangerous trees and HV lines outside ROWs | Classical methods | Uneven illumination filtering, Hough transform algorithm, motion tracking using 2D camera | [74] |
| | Detect the vegetation encroachment of transmission lines based on the image data monitored from towers with mounted binocular vision sensors | Deep learning methods | Faster R-CNN and Hough transform with advanced stereovision | [75] |
| | Estimate vegetation profile within a buffer zone, outage risk estimation of the power grid | | Residual U-Net based on GIS data, aerial and satellite imagery | [76] |

### 3.1.1. Self-Blast Glass Insulator Location

As essential electrical insulation and conductor conjunction equipment, insulators are widely used in modern high-voltage power transmission lines. According to statistics, accidents caused by insulator fault (e.g., glass insulators self-blast) account for the highest proportion of grid system failures [70]. Thus, it is of great importance to conduct periodic abnormality detection of insulators. Typically, the self-blast glass insulator location basically consists of two steps: (1) insulator localization and (2) fault identification. First of all, one needs to localize the insulators by eliminating complicated backgrounds, dynamic view, and illumination changes of the images. Then, the conditions of the insulators need to be diagnosed to determine whether they are normal or defective.

Classical methods: As discussed above, the two-step strategy locates the insulator first and then extracts the features for further classification; that is, the classifier is used to determine whether the detected insulator is normal or self-blast. Considering the various characteristics of the image, classical statistic features (such as mean, contrast, variance, histogram, color, structure, and so on) are used for feature representation. Some works transform the image to the frequency domain or complex frequency domain using Wavelet [66] and S-Transform [65] for feature extraction. Classification algorithms such as k-means [77], support vector machine (SVM) [65], and other task-specific methods are applied for insulator self-blast recognition.

Many works have been done on insulator localization and status recognition using classical methods. Liao et al. [61] proposed a robust insulator detection algorithm based on local features and spatial orders for aerial images. First, the local features are detected, and a multi-scale and multi-feature (MSMF) descriptor is presented to represent the local features. Next, several spatial order features are obtained by training the local features. Finally, a coarse-to-fine matching strategy is utilized to eliminate the background noise and locate the insulators. Wu et al. [62] introduced a texture segmentation algorithm

for aerial insulator images, which is based on principal component analysis (PCA) and global minimization active contour model (GMAC). First, the features of insulators are extracted by the gray level co-occurrence matrix (GLCM). After dividing the extracted features into those with weaker and stronger discriminative abilities, PCA is applied to features with weaker abilities to optimize the feature representation. Finally, GMAC is utilized for insulator segmentation.

Considering the characteristics contained in images, researchers have explored color and structural features to detect self-blast glass insulators. On the basis of spatial features, Cheng et al. [63] propose a method to specifically deal with the defect detection problems of twin insulator strings. According to the particular color features of glass insulators, the images are segmented, and the main axes of the insulator strings are adjusted to the horizontal direction. After marking the connected regions of insulators, spatial features including the vertical lengths of the regions, the number of insulator pixels in the regions, and the horizontal distances between two adjacent connected regions are selected. In the end, insulators with defects are located and detected based on the spatial features. However, when the insulator area is severely obstructed, it is impossible to obtain an independent connected region based on the insulator umbrella skirt, leading to poor performance of the method.

In addition to the self-contained attributes of images, several other characteristics, such as histogram, can also be explored for self-blast glass insulator location. Huang et al. [64] proposed an approach to locate and identify the self-shattering of glass insulators based on local binary pattern histograms. This method also follows the location and detection strategy. In order to reduce the interference of illumination, the color space is converted from RGB to HSI. Next, insulator strings are separated from the complex backgrounds, after segmenting the image using the maximum interclass variance method (OTSU). Then, local binary pattern histograms on sliding blocks are employed for self-blast detection.

Instead of solving the problem in the original image domain, the digital image processing (DIP) technique can be utilized to tackle the self-blast glass insulator location from another domain. Reddy et al. [65] transform the original image to the complex frequency domain using discrete orthogonal S transform (DOST). The pixel classification technique and k-means clustering are used to segment the image into various clusters first, and then a bounding box is drawn to detect the insulator. After cropping the original image along the edge of each box, DOST is applied, and features are extracted in the complex frequency domain. Finally, SVM and an adaptive neuro-fuzzy inference system (ANFIS) is applied to determine the condition of the insulator. The complex frequency domain transformation could bring extra computation complexity, leading to high computation costs in real-world situations. In addition to DOST, wavelets analysis combined with SVM [66] or hidden Markov model (HMM) [67] can also be applied for insulator condition analysis.

Though classical methods have achieved great self-blast glass insulator detection results, these methods rely heavily on prior knowledge, such as the insulator shape, size, and background. However, in the real world, insulators are quite different and their backgrounds vary significantly, from farmlands to vegetation, rivers, mountains, and towers. In addition, the hand-crafted features are low level and limited, so they are not capable of dealing with various situations.

Deep Learning Methods: Recent years have witnessed the remarkable development of deep learning methods, especially convolutional neural networks (CNN), in computer vision. Compared with hand-crafted features, one can extract more discriminative, generalized, and robust high-level feature representations using CNN. Driven by the large-scale image data of smart grids, researchers have started to apply deep learning methods to self-blast glass insulator identification tasks.

The first example of applying CNN architecture to the detection of power line insulator defects was proposed by Zhao et al. [46] and is composed of three steps. The first step is insulator detection, aiming to localize insulators with various angles. The orientation angle detection and binary shape prior knowledge (OAD-BSPK) algorithm is applied in

this step. The second step is deep feature extraction. An AlexNet [78] network pre-trained on ImageNet is employed to extract multi-patch features so as to obtain features with more rotation invariance. Finally, instead of directly using softmax, they use a one-vs.-all support vector machine (SVM) with RBF kernel to recognize insulator status.

With the emergence of new network architectures in image classification and object detection tasks, more methods have been proposed for insulator inspection. Ling et al. [61] proposed an end-to-end real-time method of self-blast glass insulator location based on faster R-CNN and U-net [68]. As an accurate and fast object detection approach, faster R-CNN is employed to localize the insulator string. After object detection, the detected insulator sting is cropped from the raw image. By this means, the low signal-to-noise ratio (SNR) of the original image is enhanced. Next, the segmented images are subjected to pixel binary classification through U-net so as to identify the broken parts. Adapted from the faster R-CNN, the mask R-CNN further develops its architecture and can realize object instance segmentation on the pixel level. Yang et al. [69] proposed a mask region convolutional neural network based on mask R-CNN to detect insulator self-shattering. This approach can locate fault insulators while simultaneously distinguishing between normal and self-blast insulators. In comparison with Ling's method, a mask region convolutional neural network can achieve better performance.

The performance of deep learning methods depends crucially on the scale of the dataset. To tackle the scarcity problem of defect images in a real inspection environment, Tao et al. [70] introduced a four-operation data augmentation method, which consists of affine transformation, insulator segmentation and background fusion, Gaussian blur, and brightness transformation. A cascaded CNN architecture is proposed to detect power line insulator defects. Following the two-level strategy mentioned above, the cascading network uses a CNN based on a region proposal network to locate the insulator position first and then detect the defect position on the insulator.

### 3.1.2. Icing Detection and Measurement

Extreme weather conditions such as snowstorms and hail disasters will harm the power transmission line. The thick icing accumulated on the transmission lines will increase their load and cause power lines to break, resulting in power supply interruption [72]. Therefore, icing detection and measurement is a crucial task in smart grids, which can help grid companies to take measures in advance for icing accident prevention.

Inspired by the traditional DIP technique, Zhai et al. [71] analyzed three kinds of edge detection method (Canny operator, Sobel operator, and adaptive weighted Sobel operator) on the UAV transmission line icing monitoring task. It is shown that the adaptive weighted Sobel operator has higher accuracy in detecting the image's edges than the other two operators. Zhong et al. [72] proposed an algorithm to automatically detect icing and estimate the ice thickness of the transmission lines. First, a calibration method calibrates the monitoring camera using just one image. An ice-prior-based scheme is then employed to accurately detect the outlines of power lines and icing. After calculating the pixel-level thickness of the lines and the icing regions based on the extracted outlines, the icing thickness can be derived from the calibration parameters and a region-of-interest (ROI) tracking scheme. In addition to just utilizing images, Huang et al. [73] also measured the distance and level angle of a target using a laser range finder and inertial measurement unit (IMU), and then realized icing parameters measurement with a photogrammetry method.

### 3.1.3. Vegetation Encroachment Monitoring

Each growing season, the overgrown vegetation encroaching on high-voltage transmission lines can cause severe blackouts/flashovers because of the short circuit faults among the conductors, posing a major threat to the security and stability of smart grids [74,75]. In these circumstances, it is critical to monitor the vegetation encroachment so as to inspect potential circuit failures of the transmission lines. In the case of vegetation encroachment

monitoring, the tasks mainly include detecting and classifying vegetation that is near the power lines and estimating its height and relative distance to the transmission lines [47].

Ahmad et al. [74] proposed a method for vegetation encroachment monitoring using a 2D camera integrated on transmission poles. With image frames acquired from cameras, the algorithm is first initialized by reference and scene images. Then, they filter out images with uneven illumination caused by rain, foggy weather, and so on. After that, the Hough transform algorithm is applied to recognize the horizontal and vertical lines far away from transmission poles. In the end, they use motion tracking to detect the encroached vegetation in the current scene image relative to reference images.

As deep learning techniques have achieved tremendous success on image recognition, several deep-learning-based detection frameworks have been proposed to monitor vegetation encroachment. Based on faster R-CNN, Rong et al. [75] proposed a detection framework with advanced stereo vision. First, they use faster R-CNN and Hough transform to detect the vegetation regions and power lines, respectively. Next, in order to obtain the precise geographical locations of the vegetation and power lines, an advanced stereovision (SV) algorithm is applied to convert the detected two-dimensional (2D) image data into three-dimensional (3D) height and location information for vegetation encroachment detection. Combining GIS data and aerial and satellite imagery, Jain et al. [76] introduced a vegetation analysis system to estimate the vegetation profile within a buffer zone. The residual U-Net architecture from [79] is utilized for vegetation segmentation.

To conclude, visible light, especially the aerial image, is an accessible and effective data source modality for abnormality detection in smart grids. With the help of conventional image processing and novel deep learning techniques, abnormality detection based on visible light has shown promising potential in smart grids, such as self-blast glass insulator location, icing detection and measurement, vegetation encroachment monitoring, and so on.

### 3.2. Infrared Image

3.2.1. Infrared-Image-Based Anomaly Detection for Low-Voltage Pulse Technology in Smart Grids

During the operation of power grid equipment, whether the condition of transformer winding in a strong electronic circuit is stable and whether the insulated current transformer works normally are very important for the monitoring of the local/general power systems [80]. Therefore, based on such a starting point, Alexander., Yu., Khrennikov., et al. [23] proposed using acoustic sensors and infrared sensors to measure the acoustic activity data of local discharge in the transformer, the air pressure data in the shell, and the internal circuit data of electronics and electricity. The data obtained by analyzing these collected data are used to detect the contact of a disconnector, and whether there are abnormalities in power transformer bushing and circuit breaker equipment [24,25]. Because it involves various electronic circuit components (such as a limiter, coupling transformer, measuring current transformer, and measuring voltage transformer), the circuit designed for low-voltage pulse is sensitive to the data changes of these electronic components. Therefore, we use the low-voltage pulse test method combined with short-circuit-induced reactance to detect abnormalities in the windings of radial and axial transformers and the currents of insulated current transformers.

As shown in Figure 1, by using infrared sensors, one can control and effectively detect the fault abnormalities of electrical equipment (e.g., overheating of disconnector contact, bushing condition of power transformer, circuit breaker condition, coupling transformer, measuring current transformer, measuring voltage transformer). The low-voltage impulse test is a very sensitive and reliable method to detect the deformation of transformer winding. The oscillogram obtained by low-voltage pulse detection is a description of the working state of the transformer winding. The main resonance frequency reflects whether the transformer winding is normal or not. The frequency spectrum of 250 MVA 220 kV winding transformer includes 110 kHz, 320 kHz, and 550 kHz, which change approximately twice

after mechanical radial winding deformation. The frequency spectrum changes after a short circuit of an 80 MVA 110 kV transformer. The original 300 kHz, 500 kHz, and 700 kHz resonant frequencies disappear, and the new 400 kHz and 800 kHz resonant frequencies disappear. Comprehensive inspection results show that the basic causes of instrument transformer failure are low quality of sulfur hexafluoride, design defects of the transformer, lack of absorber for absorbing moisture, decomposition products of sulfur hexafluoride, use of non-corrosion-resistant materials in transformer manufacturing, and low-service-quality maintenance. The quality of sulfur hexafluoride must comply with the standards specified in IEC 60480-2004. If there are defects in the hardware equipment, there will be an abnormal temperature in a part of the power transformer and insulated current and voltage transformer. This phenomenon is detected by the infrared sensor. The detected abnormal equipment will then be repaired. A summary of the above analysis is given in Table 2, with which one can find *con* and *pro* of investigated technologies.

**Table 2.** In view of the abnormal phenomena in the power grid environment, we use the following three kinds of abnormal detection technologies and methods.

| Method Category | Scenario of Anomaly Detection | Advantage | Shortcoming |
|---|---|---|---|
| Low-Voltage Pulse | Abnormal components of circuit | High precision, low energy consumption | Danger, sensitive to voltage fluctuations |
| Intelligent Sensor | Conductor defect | High reliability high stability | High power consumption |
| Internet of Things | Abnormal leakage of power grid system | Security, fast data processing | Low resolution |

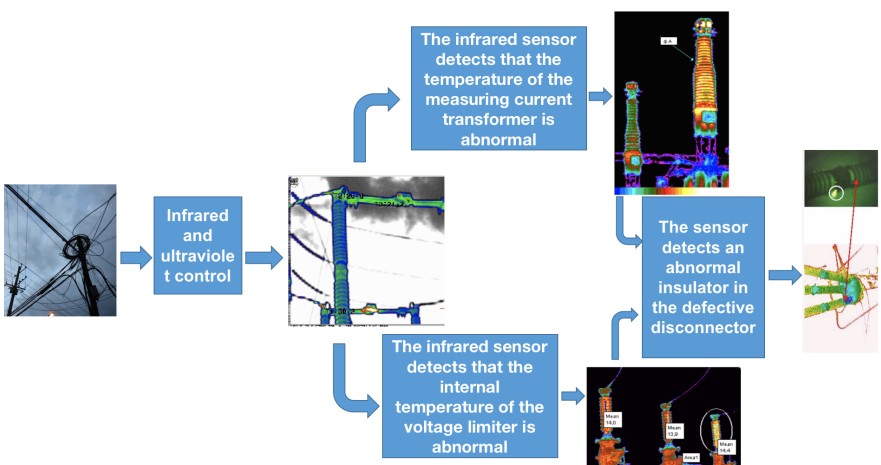

**Figure 1.** The abnormality detection of power grid insulators using low-voltage pulse and infrared sensing detection technology.

### 3.2.2. Infrared-Image-Based Anomaly Detection for Intelligent Sensor Technology in Smart Grids

Due to the wide distribution of power grid equipment, we need to carry out abnormality monitoring for each centralized power grid environment. Therefore, Semedo et al. [26] proposed using a set of integrated miniaturized intelligent detection equipment to collect data from a centralized power grid environment, using an autonomous and wireless multivariable intelligent sensor, which can collect and verify the temperature data in the high-voltage disconnector, and transmit the data obtained by the sensor to the instrument system of the remote terminal for analysis, identifying the cause of the temperature in the high-voltage disconnector. This method can successfully detect the abnormal temperature of the high-voltage disconnector in the distribution substation in a smart grid and accurately locate the abnormal temperature position.

The concepts and tools of power management and energy efficiency of the smart grid have been widely studied and promoted [22,81]. Therefore, in the process of distributing information to the whole power grid infrastructure, multiple variables describing the equipment operation status and conditions are monitored in real time. These variables are very important for the operation of each process, and perfect operation and maintenance standards should be followed, so as to better follow the standards within the power grid operation environment. Figure 2 shows the position of a remotely operated high-voltage disconnector where an abnormality has been detected using an autonomous, wireless multivariable intelligent sensor. Significant advances in different technical fields such as sensors [82], microsystems, wireless networks, and network service programming make it possible to improve this method to a higher level by combining maintenance, operation, and engineering, through the whole life cycle of assets in the power grid [83]. Specifically, some applications have solved the problem of real-time monitoring of assets, including transformers, circuit breakers, and overhead power lines [84–86]. In addition, the use of energy collection to power wireless sensor nodes in utilities is also reported [87,88]. However, the common status of remotely operated high-voltage disconnectors in primary distribution substations is indirectly evaluated by monitoring their respective motor operating mechanisms [8]. In fact, this method only reports motor drive, so it cannot provide confirmation of the forward movement of the actual disconnect switch, let alone ensure that the closed position is correct (i.e., the two components are correctly aligned). To meet this functional gap, remote operation requires one to actively confirm the actions taken in response to the appropriate commands issued by the dispatching room. The intelligent sensor introduced here actually evaluates the alignment degree of the two parts of high-voltage disconnectors through local measurement. In addition, when the circuit breaker is closed, continuous condition monitoring is very important to detect abnormal bus heating caused by poor contact in real time, to avoid hot spots. Therefore, Semedo et al. [26] proposed a new solution. A set of integrated miniaturized intelligent detection equipment is used to collect data from a centralized power grid environment, and an independent and wireless multivariable intelligent sensor is used. The sensor can collect and verify the temperature data in the high-voltage disconnector and transmit the data obtained by the sensor to the instrument system of the remote terminal for analysis; that is, it can determine the working state of the high-voltage disconnector and diagnose the faults that are occurring. This autonomous, wireless multivariable intelligent sensor is directly integrated with the workshop interconnection and integrated intelligent sensor as the bottom layer of what is now called the "instrument cloud". Because the equipment is widely distributed in the power grid, we need to use the equipment to detect each centralized power grid environment [26,28–30]. We need to transmit the data obtained by the sensor to the instrument system of the remote terminal for analysis to find out the cause of the temperature in the high-voltage disconnector.

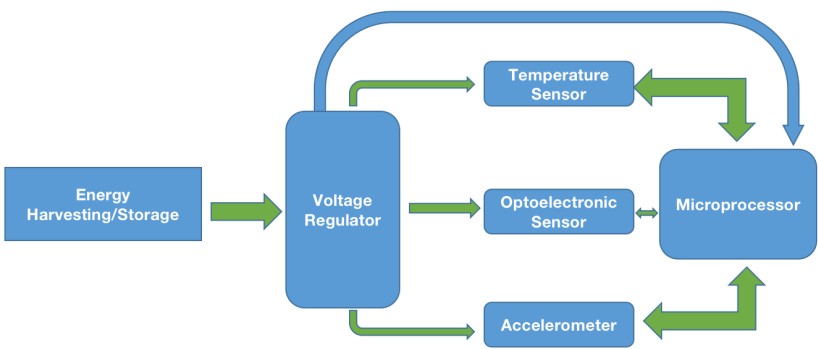

**Figure 2.** Abnormality detection with wireless multivariable intelligent sensor.

### 3.2.3. Infrared-Image-Based Anomaly Detection for Internet of Things Technology in Smart Grids

In the operation of power grid equipment, efficient energy utilization plays a vital role in the development of power systems and smart grids. Therefore, proper abnormality monitoring and control of energy consumption are two of the tasks of the smart grid. Figure 3 shows the monitoring and control of energy consumption of an electric energy meter system based on Internet of things and sensor technology. The indicator can accurately show the abnormal watt-hour meter system. There are many related problems in the existing watt-hour meter system, one of which is the lack of full-duplex communication. In order to solve this problem, an intelligent watt-hour meter [32,35–38] based on the Internet of things is proposed. The proposed smart energy meter uses a Wi-Fi module to control and calculate energy consumption and upload it to the cloud. Researchers can view the operation data of power grid equipment in the cloud. Therefore, energy analysis of the power grid environment becomes easier and more controllable. The system also helps to detect abnormal leakage of power grid equipment. Therefore, this kind of smart meter helps to realize automation and wireless communication by using the Internet of things, which is a big step towards a digital industry. The Internet of things is a network connecting intelligent devices that can transmit data. The "thing" in the Internet of things can be a person who has a heart monitor or a car equipped with built-in sensors (that is, an object that has been assigned an IP address [35,37]), and it can collect and transmit data through the network without manual assistance or intervention. Embedded technology in objects helps them interact with the internal state or external environment so as to affect decision making. In order to analyze and control power consumption, a suitable system must be established. The existing system is error prone, time consuming, and laborious. The value we get from the existing system is not accurate, although it may be digital, so the solution to this problem is to use a smart watt-hour meter. An intelligent electricity meter is a reliable device for real-time state monitoring, automatic information acquisition, user interaction, and power control. It provides a two-way information flow between the local device and the terminal and provides better controllability and efficiency. It provides real-time consumption information and energy consumption control. When the maximum load demand of the local equipment in the power grid exceeds its peak value, the power supply to the power grid equipment will be disconnected with the help of a smart watt-hour meter. In an ideal environment under a normal working load, the service life of a smart meter is about five to six years. However, in reality, intelligent electric energy meters have environmental problems, and their life is shortened with the abnormal consumption of energy. The factors affecting the service life of intelligent electric energy meters include environmental factors and varying and limited service life. The electric energy meter system based on the Internet of things mainly comprises controllers, Wi-Fi, and anti-theft detection. Whenever any fault or abnormality occurs, the abnormality detection sensor will sense the error and circuit response according to the received information.

To summarize, smart grid abnormality detection research is mainly based on the Internet of things and smart sensors. This method requires a large data transmission capacity and high precision. In addition, the sensor can be used to detect the abnormal temperature of power grid equipment and quickly check the equipment with abnormal heating. Of course, low-voltage pulses can also be used to detect abnormal voltages and currents quickly. This method has high real-time performance for detecting the internal circuits of power grid equipment.

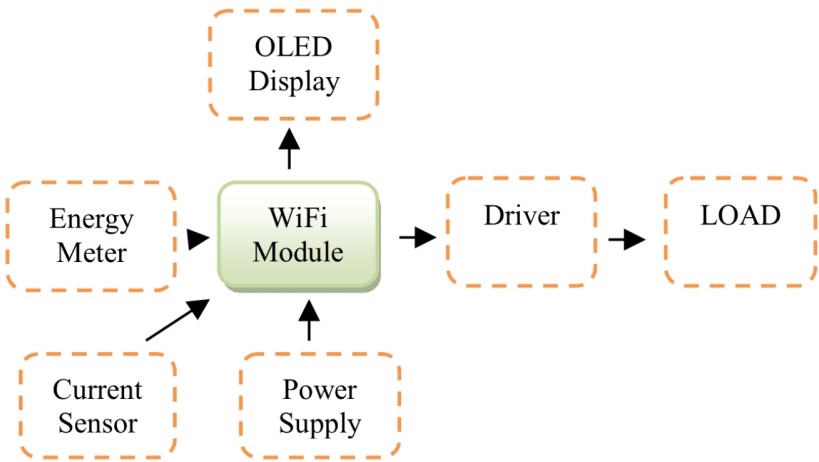

**Figure 3.** Abnormality detection in a smart energy meter system based on Internet of things and sensor technology.

## 4. Abnormality Detection with Images over Long Range

There are many forms of satellite data. For example, we can obtain positioning data through GNSS satellites, and through remote sensing satellites, we can obtain optical and SAR images. These data can provide strong support for the abnormality detection of a power grid. In this review, we will classify different types of satellite data and introduce the related work for each type of satellite data in power grid abnormality detection.

### 4.1. The Application of Satellites in Smart Grid Abnormality Detection

In this section, we introduce the abnormality detection applications in the power grid from the three aspects of satellite positioning, optical imagery, and SAR. We will summarize the abnormality detection applications suitable for each method and summarize them in Table 3.

**Table 3.** Summary of abnormality detection in power grid environment with satellite data.

| Satellite Data Types | Commonly Used Anomaly Detection Areas | Related Work |
|---|---|---|
| Satellite positioning data | Tower pole tilt detection, ground settlement detection, inspection robot detection | [10,11,89–92] |
| Optical satellite images | Vegetation monitoring | [9,12,39,93–95] |
| Synthetic aperture radar images | Tower pole displacement detection, Ice-coated area detection | [13–15] |

#### 4.1.1. Satellite Positioning Theory

GNSS is a general term for satellite navigation systems in a broad sense, including satellite positioning systems with global coverage, satellite positioning systems with regional coverage, and space-based augmentation systems. GNSS system can be divided into three main parts—space satellite, ground control station, and user terminal—according to the differences in space and function. GNSS satellites send signals containing their position information to the ground all the time. The receiver receives satellite signals on the ground, decodes the satellite ephemeris, and measures the pseudo-range, carrier phase, and other measured values. Through these measured values, the position of the user receiver can be calculated. A GNSS positioning principle diagram is shown in Figure 4.

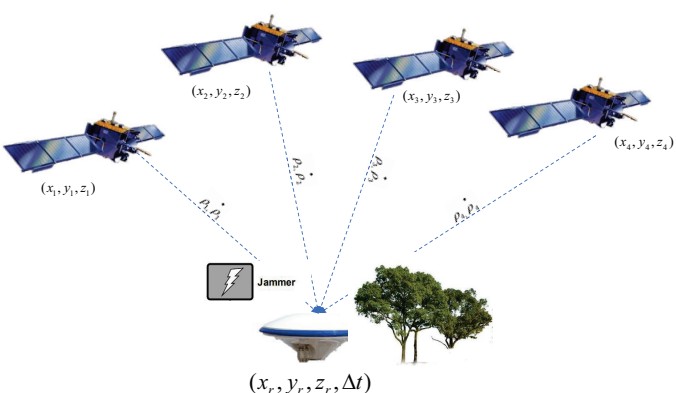

**Figure 4.** Diagram of GNSS positioning systems.

According to different measurement values and positioning methods used, satellite positioning can be divided into positioning methods such as single-point positioning, precise single-point positioning, and RTK. Although GNSS can provide high-precision services globally, the environment has a more significant impact on the positioning. The use of GNSS for abnormality detection in the power grid is a challenging problem.

### 4.1.2. Optical Satellite Images

Remote sensing usually refers to the monitoring of the earth's surface by sensing and remote sensing, and resource management (such as trees, grassland, soil, water, minerals, farm crops, fish, and wild animals) through telemetry instruments on platforms such as artificial earth satellites and airplanes. As for optical remote sensing, this mainly refers to the remote sensing technology in which the working band of the sensor is in the visible light band, that is, the range of 0.38 to 0.76 microns. With the rapid development of remote sensing detection technology, the spatial resolution, time resolution, and spectral resolution have significantly improved, providing a more accurate data source for earth observation. The spatial resolution of satellite imagery has gradually increased from the meter level to the sub-meter level. The time resolution also realizes the fastest revisit cycle once a day with the combination of constellation satellites. The continuous improvement of the resolution of remote-sensing satellites provides more reliable, more accurate, and richer data for earth observation.

### 4.1.3. Synthetic Aperture Radar Images

SAR (synthetic aperture radars) are active imaging sensors that operate in the microwave region of the electromagnetic spectrum [96]. Synthetic aperture radar is based on small antennas that move at a constant speed along the track of a long linear array and radiate coherent signals [97]; the echoes received at different positions are coherently processed to obtain higher-resolution images. It can be divided into two types: focus and non-focus. Unlike traditional optical cameras, synthetic aperture radar has an active microwave sensor, and the obtained images are not affected by the occlusion of clouds and fog, nor are they affected by light intensity and climatic factors. Therefore, they are capable of all-weather earth observation. Furthermore, based on the fact that waves encounter different materials and have different refraction and reflection characteristics, synthetic aperture radar can penetrate the ground surface and even obtain data that is hidden under the vegetation cover. These characteristics give it a wide range of application prospects in civil fields such as agriculture, forestry, water, geology, and natural disasters. It also has unique advantages in the military field, which is the basis for the use of inspection robots for abnormality detection.

### 4.2. Grid Anomaly Detection Based on Satellite Positioning

The satellite positioning described in the previous section can provide high-precision global coordinates. Therefore, in the application of power grid abnormality detection, it

is often used in power facility settlements, tower pole tilt detection, and high-precision positioning for various unmanned patrol  systems.

### 4.2.1. Intelligent Unmanned Inspection Systems

As shown in Figure 5, the use of drones has become an essential means of transmission line inspections. At the same time, there are also explorations of related applications in disaster surveys, emergency repairs, and drones to remove foreign objects in ground wires. At present, the application of drones is mainly based on the manual operation of drones for inspections. There are problems of high requirements for operators and high labor costs. Autonomous inspections by drones are the future trend. Due to safety considerations, there are high requirements for high-precision positioning and navigation performance. Therefore, there are many studies based on GNSS data to explore high-precision positioning methods for inspection drones and inspection robots.

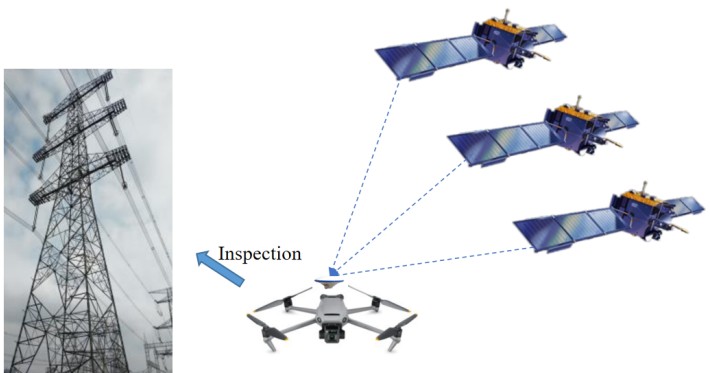

**Figure 5.** Application of satellite-positioning-and-image-based abnormality detection in power systems.

As we all know, there is substantial electromagnetic interference in the power grid environment, which will cause severe interference to inspection drones that use GNSS for positioning. Shepard et al. [10] has conducted a comprehensive evaluation UAV positioning's vulnerability to GPS spoofing attacks in the smart grid scenario. Although, with the development of satellite positioning technology, the emergence of pseudo-range differential positioning, RTK, and PPP alleviate, to some extent, the positioning problem of using GNSS in the power grid, the robustness of UAV positioning is still driving progress in this field. Therefore, in response to the GNSS challenge in the power grid scenario, ref. [11] proposed an experimental flight test evaluation of a cooperative navigation strategy in which an unmanned aerial vehicle (UAV) that is subjected to very poor GNSS satellite geometry is provided ranging updates from an unmanned ground vehicle (UGV). However, more research is required to use multi-sensor fusion to solve the problem of GNSS interference in the power grid. The most classic use of GNSS and IMU for fusion, the GNSS/INS integrated navigation system, can operate robustly in most scenarios [98]. In recent years, there have also been studies on the integration of GNSS with vision and lidar SLAM [99], which has also achieved good results and can operate stably in challenging GNSS environments such as power grids. The high-precision positioning results of GNSS are applied to the intelligent unmanned inspection system to provide protection for the abnormal detection of the power grid environment.

### 4.2.2. Tower Pole Monitoring and Settlement Detection

The high-precision positioning results of GNSS in static measurement are often used in abnormality detection tasks such as tower pole detection in the power grid. The power tube pole is a necessary facility to carry the power supply, whose free operation is of great significance to ensure the safe operation of the power grid [89]. However, traditional manual detection methods are time consuming and labor intensive and do not guarantee good real-time performance. Therefore, Gou et al. [90] designed a high-precision satellite data



acquisition system for the power grid environment. Furthermore, to solve the problems of low monitoring accuracy, high false alarm rate, untimely pre-warning, and high equipment failure rate of the transmission line tower rod oblique posture monitoring system, Yi et al. [91] proposed a transmission line tower rod oblique monitoring system based on reverse network RTK. Moreover, Shen et al. [92] proposed technology for monitoring the tower pole tilt attitude based on the RTK technology of a Beidou reverse network and measuring the tilt attitude information of a transmission line tower by using its height and position information. Then, using this system, the risk assessment is carried out, which can control the tilt attitude and assess the risk of the transmission line tower.

### 4.3. Grid Anomaly Detection Based on Optical Satellite Images

In order to deal with the potential safety hazards that high vegetation may cause to transmission lines, Moeller et al. [9] discussed a new method based on the analysis of multi-spectral and stereoscopic very-high-resolution satellite imagery, which contains three steps: (1) extract the tall vegetation from the multi-spectral bands; (2) use stereo images to evaluate the vegetation height; (3) map the vegetation with a potential for interference. Kobayashi et al. [93] introduced a concept for the use of multi-spectral stereo pairs of satellite images to identify dangerous trees and plants along with overhead transmission rights of way, in which the height of each pixel is determined, and the distance from the conductor is calculated with the multi-spectral stereo images. Ahmad et al. [94] also proposed a vegetation-monitoring method with satellite stereo images. In this work, multi-spectral and panchromatic images are first merged, and then normalized difference vegetation index and stereo image matching are used to detect the vegetation and calculate the height of each tree, respectively. In recent years, some related studies have also emerged. Xiao et al. [95] proposed a novel method to use multi-view satellite images to detect individual trees and delineate their crowns. This method generates the DSM from multi-view high-resolution satellite images and combines it with spectral information to detect the trees. With the development of deep learning, various vegetation detection algorithms using machine learning methods have emerged. Gazza et al. [12] proposed an automated framework for monitoring vegetation along power lines using high-resolution satellite imagery and a semi-supervised machine learning algorithm, which effectively provided updated situational awareness to vegetation management teams in electric utilities.

### 4.4. Grid Anomaly Detection Based on SAR Images

In this section, we will introduce abnormality detection with SAR images in a smart grid environment. As a basis, Schwarz et al. [16] studied how a power line tower was visible in time-frequency decomposition results. Sha et al. [17] used a time series of SAR images and studied the backscattering behavior of a power line segment with six conductors. This also makes it possible to use SAR images to detect power grid abnormalities, such as disaster monitoring. Finally, we try to list the works using satellite SAR in power grid detection, although these works are not many. Yan et al. [18] discussed the use of SAR data to monitor power transmission towers in natural disaster conditions, by extracting the height of towers from single images and exploiting this information to detect collapsed or distorted towers. Li et al. [13] detected the frozen scene in an ultra-high power tower with satellite SAR images. They studied the ultra-high voltage (UHV) power tower target interferometry in TerraSAR Spotlight (TSX-SL)-mode SAR images and testified that the variation of the radar scattering intensity on the tower cross could be used to quantitatively evaluate towers' icing state. Similarly, some work has studied the settlement of the power grid and the movement of power towers [14,15] with satellite SAR images. Talright et al. [15] applied the persistent scatterer interferometry (PSI) method to monitor power towers' movement with the SAR data from two satellites. Based on the large-scale coverage of Sentinel-1 SLC images in IW TOPS mode, He et al. [14] utilized persistent scatterer interferometry to measure the ground target movements along transmission lines with millimeter accuracy. Their experiments showcase that the time series of transmission

tower displacement is related to the seasons, among which winter and spring are relatively quiet. These works also prove that advanced InSAR technology has broad application prospects in the realization of smart grids.

Interested readers are also referred to the summaries in Table 4, in which we provide general platforms that one can choose for one's own works. In the following, we will provide methodologies related to abnormality detection in smart grids.

**Table 4.** Platforms For collecting visible light data.

| Visible Light Data Resource | Advantages | Shortcomings |
|---|---|---|
| Ground patrol | High detection rate | Slow, tedious, dangerous for human, labor-intensive, impossible in extreme weather conditions and harsh terrains |
| Fixed camera | Low maintenance | Limited observation range, complex deployment, high cost |
| Climbing robot | High inspection accuracy | Slow, weight could damage the lines, hard to pass across various obstacles |
| Helicopter | Fast inspection speed, wide observation range | High cost, safety issues |
| Unmanned Aerial Vehicle (UAV) | Automatic or human-controlled, real-time, safe, fly close to the detection target | Difficult for power line tracking and navigation |

## 5. Methodologies

Abnormality detection methods can be generally categorized into three classes: model-driven [4,5], data-driven [6,7,40,44], and machine-learning-based [45–47]. We will provide a very brief introduction to the first two categories and put more focus on the third, as this survey focuses on abnormality detection with image data, in which the deep-learning-based solutions will play a more important role in data modeling and solutions.

### 5.1. Abnormality Detection with Model-Driven Methods

Abnormality detection with model-driven methods is a well-established research areas, which adopts many advanced signal-processing techniques [5,100] to get the indicators. To deliver an effective survey, we will start by introducing the well-known linearized power flow equations [5], which can be expressed as follows:

$$\left[ \begin{array}{c} \Delta \mathbf{F} \\ \Delta \mathbf{G} \end{array} \right] = \left[ \begin{array}{cc} \mathbf{F}_\theta & \mathbf{F}_\mathbf{v} \\ \mathbf{G}_\theta & \mathbf{G}_\mathbf{v} \end{array} \right] \left[ \begin{array}{c} \Delta \theta \\ \Delta \mathbf{v} \end{array} \right] = \mathbf{J} \left[ \begin{array}{c} \Delta \theta \\ \Delta \mathbf{v} \end{array} \right], \tag{1}$$

where $\mathbf{v}, \theta$ represent the voltage magnitude and node angles, respectively. It is well accepted that the minimum singular value of the power-flow Jacobian matrix $\mathbf{J}$ can be trusted as a static abnormality indicator [5]. Equation (1) can be rewritten as follows:

$$\mathbf{A} = \mathbf{B}^T \mathbf{C}, \tag{2}$$

where $\mathbf{A} = \left[ \begin{array}{cc} \Delta \theta & \Delta \mathbf{v} \end{array} \right]^T \in C^{p \times N}$; $p$ and $N$ denote the number of buses and variables, respectively; $\mathbf{B}^T = \mathbf{J}^{-1}$; and $\mathbf{C} = \left[ \begin{array}{cc} \Delta \mathbf{F} & \Delta \mathbf{G} \end{array} \right]^T$.

From the above, we can see that the maximum singular value of $\mathbf{B}$ also equivalently indicates the abnormality. Interested readers are referred to the works of [4,48] for technical details. Based on Equations (1) and (2), abnormality detection with model-driven methods is widely adopted in classical power systems. However, with an increase in the size of the power system, such a method will pose more of a challenge in solving the large-scale power flow equations. More essentially, the new techniques are hardly modeled in the classical methods, which calls for new solutions for abnormality detection.

### 5.2. Data-Driven Modeling for the Power System

In the era of big data [42,101], the data-driven methods provide a new kind of solution for reliable WAMPAC. For example, Xie etc. proposed a robust principal component analysis (PCA)-based abnormal detection method [6]. The singular value decomposition

(SVD)-based indicator was advanced by the work of [40]. Inspired by random matrix theories, the linear eigenvalue statics of considered PMU data were proposed by [7]; however, this brought a massive computational expense due to the high dimensionality of the employed data. Recently, based on the high-dimensional covariance matrices test, the work of [50] proposed a low-complexity algorithm to address this challenging issue, which works well in both a small- and large-scale power system. Very recently, using random matrix theory [49,52,54,102], Yang et al. proposed an improvement of power system state estimation by presenting matrix cleaning (MC) technology [44]. Instead of numerically introducing the details of these methods, we provide insightful results below, with which one can understand the technical connections with the classical methods.

Taking the trace operator of the Equation (2), we can get

$$\mathbf{Tr}(\mathbf{A}) = \mathbf{Tr}\left(\mathbf{B}^T\mathbf{C}\right). \tag{3}$$

Denoting the normalized trace operator as $\mathbf{tr}(\cdot) = \frac{1}{N}\mathbf{Tr}(\cdot)$, then we can have

$$\frac{1}{N}\mathbf{Tr}\left(\mathbf{A}\mathbf{A}^T\right) = \frac{1}{N}\mathbf{Tr}\left(\mathbf{B}^T\mathbf{C}\mathbf{C}^T\mathbf{B}\right). \tag{4}$$

It is noted that the trace operator obeys the following property:

$$\mathbf{tr}(\mathbf{XYZ}) = \mathbf{tr}(\mathbf{ZXY}), \tag{5}$$

$$\mathbf{tr}\left(\mathbf{X}^T\mathbf{Y}\right) \leq \left[\mathbf{tr}\left(\mathbf{X}^T\mathbf{X}\right)\right]^{1/2}\left[\mathbf{tr}\left(\mathbf{Y}^T\mathbf{Y}\right)\right]^{1/2}, \tag{6}$$

Combining (4)–(6), we can obtain

$$\mathbf{tr}\left(\mathbf{A}\mathbf{A}^T\right) = \mathbf{tr}\left(\mathbf{B}^T\mathbf{C}\mathbf{C}^T\mathbf{B}\right) = \mathbf{tr}\left(\mathbf{B}\mathbf{B}^T\mathbf{C}\mathbf{C}^T\right), \tag{7}$$

and

$$\mathbf{tr}\left(\mathbf{A}\mathbf{A}^T\right) \leq \left[\mathbf{tr}\left(\mathbf{B}^T\mathbf{B}\mathbf{B}\mathbf{B}^T\right)\right]^{1/2}\left[\mathbf{tr}\left(\mathbf{C}^T\mathbf{C}\mathbf{C}\mathbf{C}^T\right)\right]^{1/2}. \tag{8}$$

Let $\sigma_{\mathbf{max}}$ be the maximum singular value of $\mathbf{B}$, and $\sigma_{\mathbf{max}}$ satisfies

$$\sigma_{\mathbf{max}}^2 \geq \mathbf{tr}\left(\mathbf{B}^T\mathbf{B}\mathbf{B}\mathbf{B}^T\right), \tag{9}$$

From (7)–(9), we can get

$$\mathbf{tr}\left(\mathbf{A}\mathbf{A}^T\right) \leq \sigma_{\mathbf{max}}\left[\mathbf{tr}\left(\mathbf{C}^T\mathbf{C}\mathbf{C}\mathbf{C}^T\right)\right]^{1/2}. \tag{10}$$

Base on the analysis from (1) and (10), the normalized trace operator of $\mathbf{A}\mathbf{A}^T$ can be equivalently regarded as a novel indicator of abnormality detection in smart grids. The above simple analysis can bring critical insight to the basics of data-driven methods.

In conclusion, data-driven abnormal detection can provide reasonable solutions with high-dimensional data [44,50,103,104]. In addition, they need no information of the power parameters or network topology [7,42,101]. However, in terms of the availability of data, image data pave a new means for abnormality detection problems, with which one can achieve the goal of abnormality detection in the air and over the air (referred to as small-distance and large-distance, respectively). The following section will provide an overview of deep learning methods, which are key enablers for image-data-based abnormality detection.

### 5.3. Abnormality Detection with Deep Learning Methods

Deep learning (DL) has gained a lot of popularity in recent years since it significantly improves the ability to learn the input data in their raw form and extract the hidden information with multiple levels of abstraction [23,105,106]. It has been widely used in various

applications, such as natural language processing [26], visual image classification [32], object detection [107], and activity recognition [108–112]. Anomaly detection is a key enabler that helps find potential damage in real time and recover from failure as soon as possible. DL also plays a pivotal role in abnormality detection. The performance of conventional machine learning methods cannot be guaranteed since it is challenging to deal with large-scale data and sequential data. Furthermore, the boundary between normal and anomaly elements is usually not well defined, and sometimes new types of faults may occur. These reasons make DL a superior solution in separating abnormalities from normal operations. From the perspective of mathematical modeling, deep-learning-based abnormality detection mainly depends on the following hypothesis:

$$\begin{cases} H_0: & I(\mathbf{x}) \leq \delta \\ H_1: & other \end{cases} \tag{11}$$

where $H_0$ and $H_1$ are related to normality and abnormality, respectively; $I$ denotes the indicator learning by a neural network, which takes the image data $\mathbf{x}$ as input; and $\delta$ is the threshold. One can also formulate the above hypothesis as a binary classification problem.

According to the availability of labeled data, deep-learning-based abnormality detection models can be also classified into three main categories: supervised learning, semi-supervised learning, and unsupervised learning. The most common one in this field is supervised learning. The approach of supervised learning is very intuitive. Both normal and abnormal data instances are provided with correct labels to train the deep learning model. However, a large amount of labeled data is necessary for supervised deep abnormality detection, which makes it very time consuming to collect enough data. Furthermore, anomaly events are rare, so they are more difficult to capture. Even though the supervised learning approach is able to achieve a better performance [113], the lack of labeled data and the class imbalance are two major challenges. On the other hand, semi-supervised learning only needs some labeled data, and unsupervised learning can find the intrinsic information from unlabeled data, so it can deal with the challenges mentioned above. Therefore, many investigations have been done regarding the semi-supervised learning and unsupervised learning methods [24]. A concise summary of the three approaches is shown in Figure 6. In the following sections, we introduce and discuss some works about supervised learning, semi-supervised learning, and unsupervised learning, respectively.

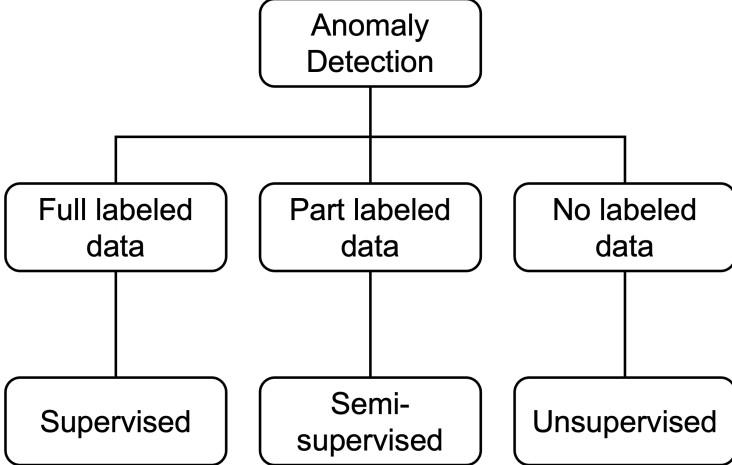

**Figure 6.** Deep-learning-based anomaly detection methods.

5.3.1. Supervised Learning

The deep supervised learning method focuses on training a classifier to separate the data classes. A convolutional neural network(CNN) is one of the popular solutions. In general, a CNN consists of a series of feed-forward layers, which are the input layer, convolutional layer, pooling layer, fully connected layer, dropout layer, and activation layer.

Because of these layers, the CNN is able to handle large-scale data and achieve a good performance in classification tasks. CNN has also often been adopted to detect anomalies. Li et al. proposed a CNN model together with random forest to detect electricity theft in a power grid system [27]. The model separates the non-technical losses from technical losses, and it is evaluated with real energy consumption data. The results show an advancement in terms of accuracy and efficiency. In [33], the authors adopted two CNNs in parallel to detect pavement cracks, and they were the first to do so. Two databases with different scales of grid are provided as the training sets, and each CNN is trained by one of the two databases. During the test phase, the results from both CNNs are compared, and the grids containing cracks are kept. Then, principal component analysis (PCA) is applied after the CNN to estimate the crack's type, classifying it as a longitudinal crack, transverse crack, or alligator crack. The model achieves more than 90% accuracy for all three crack types.

Although CNNs have remarkable performance with large-scale input data, they cannot deal with temporal information. Recurrent neural networks (RNNs) address this problem with loops that preserve and learn from past instances. However, traditional RNNs suffer from gradient vanishing and exploding problems. A modified version called a long short-term memory (LSTM) network was invented to resolve these problems [34]. An LSTM network with stacked architecture was also proposed to detect anomalies in time series data [28]; the evaluation was performed with four different datasets for different purposes, and the effectiveness of the LSTM was verified. In [29], a supervised LSTM network was applied to determine non-robust statistical properties. Then, it was combined with normal statistical analysis to detect outliers, and the approach was evaluated in terms of precision, recall, and F-measure. He et al. adopted an LSTM network to detect zero-day attacks against the controller code in a power grid system [19]. The network was first trained with normal hardware readings from performance counters, and then it estimated the distribution deviations from the normal behavior with an effective statistical test. The method was tested with six zero-day attacks and reached a detection rate near 100%, with latency less than 360 ms.

Besides the aforementioned methods, a hybrid method that combines CNN and LSTM has been proposed in [25]. The method takes both data measurements and network-level features as input to estimate the system states jointly, and false data injection attacks in the power grid system can be detected in real time. The experiment shows superiority compared to traditional state estimation methods.

### 5.3.2. Semi-Supervised Learning

Semi-supervised learning is an effective approach when unlabeled data is used in conjunction with a smaller amount of labeled data during the training phase. It is valuable in abnormality detection since, in some cases, the labeled data for abnormal instances is absent, or the classifier has to distinguish new types of anomaly [114]. Like supervised learning, CNN and RNN networks also work well with some variations. Perera et al. proposed a CNN-based method to identify an abnormal instance, with only one normal class used in training [20]. Instead of the traditional classification loss function, compactness loss and descriptiveness loss functions were proposed and applied together with a parallel architecture. The method was tested on a public abnormality detection dataset and had a better performance than some other methods. A 3D CNN network was presented to improve the localization of extreme weather events. The temporal information and unlabeled data were also taken into account to have more accurate bounding box predictions compared to the 2D CNN network [35]. In [115], an RNN-based method was proposed to classify hyperspectral images with just a small amount of labeled data, while the unlabeled training data was adopted with pseudo-labels generated by a non-parametric Bayesian clustering algorithm.

AutoEncoder (AE) is an efficient approach to learning both linear and nonlinear features from unlabeled data, so it is widely used in semi-supervised and unsupervised learning [36]. Edmunds et al. applied AE together with a neural network to detect a

dynamically targeted anomaly, and it achieved a good performance in finding anomalies from majority classes [21]. Deep AE has also been adopted in finding implausible data from electronic health records (EHR) [116–118], which can reduce the workload of doctors [37]. The method was evaluated with multivariate data from 30 clinical observations.

Furthermore, generative adversarial networks (GAN) have been used to detect anomalies that are absent in the training phase [38]. GANs also show success in generating training data for classifiers [86]. A multi-class classifier is trained with nominal and novel data that is generated from a generator, and the generator is trained with feature-matching loss in order to cheat the classifier. In [30], the authors proposed a corrupted GAN, which does not need convergence during training and addresses the one-class classification task. The method had a better performance than a solution based on AE.

### 5.3.3. Unsupervised Learning

Unsupervised learning is essential in abnormality detection since the labeled data is not always available. It is possible to find the inherent boundary that separates the normalities and abnormalities. AE is the fundamental architecture for unsupervised learning. A relational AE was proposed that shows a strong ability to extract features in high-dimensional data, and a low classification error rate was obtained with well-known datasets [83]. Abati et al. designed a deep-AE-based method, together with a parametric density estimator, to learn the probability distribution [22]. The experiments were performed with three data types: images, videos, and cognitive data. The performance showed the method is transferable to diverse contexts. A mixture method was proposed in [81]. Deep AE first reduces the dimensions and feeds the extracted features into a Gaussian mixture model (GMM). Unlike other cascade architectures, the parameters of AE and GMM are shared and optimized in a joint way, which solves the inconsistent optimization problem in decoupled models.

To process time-series data, RNNs are also widely used in an unsupervised way. In [84], RNN was adopted for early detection of cyber attacks, and it had a better performance than dynamic PCA. Singh proposed an LSTM method for abnormality detection [85]. The trade-off between prediction accuracy and abnormality detection ability was discussed. A better setting for prediction may not be a good one for abnormality detection. The evaluation was done with three real-world datasets and shows that the method works well for both time series modeling and abnormality detection. Lawson et al. proposed the first GAN-based method to find anomalies for a patrolbot [87]. In each monitoring environment, a separate GAN is set up to learn normal objects in an unsupervised way. To make real-time application possible, a sliding window with a small fixed length is used. The evaluation is carried out with a mobile robot that projects circular images, and a low false-positive rate is achieved.

In this subsection, we discussed some deep learning techniques used in abnormality detection. Based on the availability of labeled data, supervised learning, semi-supervised learning, and unsupervised learning were discussed and presented. Table 5 summarizes the related work in terms of the techniques used. Various approaches have been proved to be effective in different detection scenarios. The goal of this subsection was to give a comprehensive view of deep learning model selection for a particular domain or dataset from power grids. It is noted that we also include some state-of-the-art deep-learning-based solutions that go beyond the data collected in smart grids, which may help interested readers build their own network models for their complex issues.

### 5.4. Discussion

This subsection discusses the pros and cons of deep-learning-based abnormality detection vs. classical machine-learning-based ones, (i.e., SVM, random forest, and their variants). As shown in Table 6, both classical ML and DL models are data-driven, and both employ deterministic and stochastic optimization strategies. However, a DL with some customized designs (denoising module) is more robust than an ML in the case of noisy data.

The ML models can be more easily trained when compared to DL with a large number of neurons.

**Table 5.** Deep-learning-based anomaly detection techniques.

| Technique Used | References | Features |
|---|---|---|
| Convolutional neural network (CNN) | Supervised: [27,33]<br>Semi-supervised: [20,35] | Ability to deal with large-scale data |
| Recurrent neural network (RNN)<br>Long short-term memory (LSTM) | Supervised: [19,28,29]<br>Semi-supervised: [115]<br>Unsupervised: [84,85] | Processing time series data |
| AutoEncoder (AE) | Semi-supervised: [21,37]<br>Unsupervised: [22,81,83] | Simple, ability to learn from unlabeled data |
| Generative adversarial networks (GAN) | Semi-supervised: [30,38,86]<br>Unsupervised: [87] | Generate plausible data to boost classifier |

**Table 6.** Comparison of ML- and DL-based abnormality detection.

| | Highlights | Disadvantages | Usage and Applicability |
|---|---|---|---|
| Classical ML abnormality detection | (1) Data-driven;<br>(2) Deterministic optimization strategy;<br>(3) Performance may saturate with a large amount of data. | (1) High data/model error susceptibility;<br>(2) Not robust to noisy data. | (1) Commonly used in lightweight devices;<br>(2) Well-studied methods. |
| DL-based abnormality detection | (1) Data-driven;<br>(2) Scholastic gradient descent;<br>(3) Data hungry;<br>(4) Efficient inference with well-trained model. | (1) Data and label hungry;<br>(2) High storage and computation requirement;<br>(3) Model interpretability is needed.<br>(4) Complex algorithm selection process; | (1) Mostly used in well-equipped platforms;<br>(2) Widely studied, but not well studied. |

It is noted that the performance of both these two kinds of models will increase as the availability of labeled data increases. However, the performance of ML-based abnormality methods may become saturated with a large amount of data, while the DL models can harvest more benefits from a massive dataset. Moreover, the DL models are sensitive to the existing volume of data, and their performance may sharply decrease in the case of data scarcity. Fortunately, there exist many methods to overcome this challenging issue. For example, one can employ data augmentation methods to increase the amount of data [119]. In addition, one can apply advanced mathematical principles to obtain data-efficient DL models [120]. Moreover, employing multimodal data [121] (i.e., various types of image data in this survey) can further overcome the challenge of a shortage of data. We conclude this subsection by pointing out that we are a long way from achieving the goal of ultra-reliable abnormality detection based only on time-series or image data. In fact, abnormality detection with multimodal image data is an essential yet challenging research direction worthy of in-depth exploitation.

## 6. Conclusions and Future Work

This paper surveys state-of-the-art image-data-based abnormality detection in smart grids and outlines several key enablers. Considering the intricacies of the applications, we have broadly divided this survey into two parts: image-data-based abnormality detection and related methodologies. Within each of these topics, we have mainly surveyed the various approaches that have been advanced for enabling smart grids to run intelligently. Nevertheless, considering that the applications of image data in abnormality detection are far from well established, there are quite a large number of issues that require in-depth investigation. For example, the deployment of equipment for collection of high-quality image data and solutions for all-day environments are expected to facilitate possible future research on multimodal image-data-based abnormality detection at long range.

**Author Contributions:** Conceptualization, G.W., R.H., L.P., W.Y., L.C. and L.P.; Formal analysis, L.C. and L.P.; Funding acquisition, L.P., W.Y. and R.Q.; Investigation, G.W., R.H., L.P., W.Y., L.C. and L.P.; Methodology, F.Z., G.W., Y.M., R.H. and L.C.; Project administration, Y.M., H.G., R.H., L.P. and R.Q.; Resources, R.Q., L.P. and W.Y.; Supervision, R.Q., L.P. and W.Y.; Visualization, G.W., R.H., L.P. and W.Y.; Writing—original draft, All authors; Writing—review & editing, All authors. All authors have read and agreed to the published version of the manuscript.

**Funding:** This research was funded by the project No. YNKJXM20191246 and N.S.F.C 61873163.

**Institutional Review Board Statement:** Not applicable.

**Informed Consent Statement:** Not applicable.

**Acknowledgments:** The authors thank the editor and reviewers for their useful comments, which have improved the quality of this paper. The authors acknowledge the funding supporting the project (No. YNKJXM20191246), which focuses on the construction of satellite remote-sensing technology for power applications and wide-area intelligent monitoring of environments. The authors also thank members from the MSP group at Shanghai Jiao Tong University for their discussions and proofreading of this paper.

**Conflicts of Interest:** The authors declare no conflict of interest.

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
