# Peer review of "A Comprehensive Survey for Deep-Learning-Based Abnormality Detection in Smart Grids with Multimodal Image Data"

_applsci, doi:10.3390/app12115336_

Round 1

Reviewer 1 Report

Authors did comprehensive survey for the title " A Comprehensive Survey for Deep Learning based Abnormality Detection in Smart Grids with Multi-Modality Imaginary Data".Manuscript is well prepared,however certain queries need to be clarified from authors which are as follows :

  1. Why authors selected only deep learning algorithms in study.Why not other machine learning models like SVM,ANN,Random forest etc.
  2. Authors should clearly define the advantages of deep learning algorithms over traditional ML algorithms.
  3. Do there is a significant role of image processing techniques in  Abnormality Detection in Smart Grids.If yes kindly include a section in revised manuscript.
  4. When the amount of dataset is low,generally deep learning algorithms are not able to predict the abnormality with high accuracy.How this issue can be addressed.

Author Response

Please see the attachment for the point-to-point responses for your comments. Thank you!

Reviewer 2 Report

  • The paper an appropriate length.
  • The key messages short, accurate and clear.
  • The text’s meaning is clear.
  • A well-written the introduction.
  • Sets out the argument ....
  • Summarizes recent research related to the topic ...
  • Highlights gaps in current understanding or conflicts in current knowledge ...
  • Establishes the originality of the research aims by demonstrating the need for investigations in the topic area.
  • Original and topicality can only be established in the light of recent authoritative research.
  • This research has enough data points to make sure the data are reliable.
  • The results seem plausible, the trends you can see support the paper's discussion and conclusions, There are sufficient data.
  • The references relevant, recent, and readily retrievable.
  • The author has a deep understanding of the paper's content.
  • The research has most interesting data.
  • Abstract highlight the important findings of the study.
  • The authors presenting findings that challenge current thinking.
  • The evidence they present strong enough to prove their case.
  • The correct references cited.

Author Response

(The authors gave the same response as above.)

Reviewer 3 Report

In the paper, the authors have surveyed the recent advances in abnormality detection in smart grids with multi-modality imaginary data, which include visible light, Infrared, and Optical Satellite Images. For each of these topics, they had mainly surveyed the various approaches that have been advanced for enabling smart grids to run intelligently. However, there are some observations: 
1.    The introduction should be extended with more details on “abnormality Detection in Smart Grids” to highlight better the importance of this topic. The authors should provide a more sufficient critical literature review to indicate the drawbacks of existing approaches compared with the proposed approach in the paper. The authors should highlight the novelty of the proposed topic. The readers need more convinced literature reviews to indicate the main contributions.
2.    The authors assert that “Recently, academic researchers and the power companies tried to improve the wide-area monitoring, protection, and control (WAMPAC) [1–4] by employing a huge number of PMUs, which enables robust and accurate power state management [5,6].” But, 3 references ([4], [5], and [6]) are old, from 2000, 2009, and 2013, and cannot be considered recent.
3.    Additional information should be introduced for each statement. “For example, by the year 20 2019, more than 2400 PMUs were stationed [2].” Where were these 2400 PMUs stationed?
4.    Section 2: For the aspects “On the Visible Light Images” the references are missing. 
5.    Section 2, Rows 54 – 55: “We divide the power grid anomaly detection methods into three categories. The three methods are as follows”. The authors should use either categories or methods. A category can have more methods.
6.    Table 2 exceeds the page edge. Please verify.
7.    The explanations in Figures 1 -5 should be presented in the text, and the titles should be clear.
8.    Section 5: Why was given only for the first two categories of mathematical modeling and not for all three?  It would be useful for readers to provide details for each category. 

Author Response

(The authors gave the same response as above.)

Round 2

Reviewer 3 Report

The authors have tried and largely succeeded in removing any doubts raised by the reviewer.